# Rush Hour of LATs towards Their Transport Cycle

**DOI:** 10.3390/membranes11080602

**Published:** 2021-08-08

**Authors:** Adrià Nicolàs-Aragó, Joana Fort, Manuel Palacín, Ekaitz Errasti-Murugarren

**Affiliations:** 1Laboratory of Amino Acid Transporters and Disease, Institute for Research in Biomedicine (IRB Barcelona), The Barcelona Institute of Science and Technology (BIST), Baldiri Reixac 10, 08028 Barcelona, Spain; adria.nicolas@irbbarcelona.org (A.N.-A.); joana.fort@irbbarcelona.org (J.F.); 2CIBERER (Centro Español en Red de Biomedicina de Enfermedades Raras), 08028 Barcelona, Spain; 3Department of Biochemistry and Molecular Biomedicine, Universitat de Barcelona, 08028 Barcelona, Spain

**Keywords:** APC, LATs, SLC7, transport cycle, structure, substrate binding, substrate translocation

## Abstract

The mammalian SLC7 family comprises the L-amino acid transporters (LATs) and the cationic amino acid transporters (CATs). The relevance of these transporters is highlighted by their involvement in several human pathologies, including inherited rare diseases and acquired diseases, such as cancer. In the last four years, several crystal or cryo-EM structures of LATs and CATs have been solved. These structures have started to fill our knowledge gap that previously was based on the structural biology of remote homologs of the amino acid–polyamine–organocation (APC) transporters. This review recovers this structural and functional information to start generating the molecular bases of the transport cycle of LATs. Special attention is given to the known transporter conformations within the transport cycle and the molecular bases for substrate interaction and translocation, including the asymmetric interaction of substrates at both sides of the plasma membrane.

## 1. Introduction

Transporters have been of paramount importance since cellular compartments appeared during the evolution of life. Therefore, well-regulated communication between the two distinct environments that inevitably emerged from this compartmentalization was fundamental. Proteins embedded in the lipid bilayers appeared later and functioned not only as transporters, but also participated in the interaction between cells and their surroundings [1]. These embedded molecules account for about 25% of all proteins encoded by the human genome [2] and around two-thirds of known druggable targets in the cell [3,4], including receptors, channels, and transporters. Membrane proteins are major pharmaceutical targets because they play key biochemical roles in cell communication, signal transduction, and the transport of molecules across membranes [5].

Amino acid availability regulates cell physiology [6]. The transfer of amino acids across the plasma membrane is mediated by specific transporter proteins that recognize, bind, and transport these molecules from the extracellular medium into the cell, or vice versa. LATs, together with CATs, comprise the SLC7 family of transporters [7], although, significant differences exist between them: (i) in metazoans, LAT transporters heterodimerize with either CD98hc (LAT1, LAT2, y + LAT1, y + LAT2, Asc-1, and xCT) or rBAT (b^0,+^AT and AGT1), forming the so-called heteromeric amino acid transporters (HATs) [7], while CATs are monomeric transporters; (ii) the number of transmembrane domains is different in LATs (12) compared to CATs (14); and (iii) LATs mediate the obligatory exchange of all amino acids (except proline), while CATs act as both exchangers and facilitators of cationic amino acids [7]. The mammalian SLC7 family belongs to the APC family, which is part of the APC superfamily [8]. Several human pathologies highlight the physiological roles of SLC7 transporters [9]. Indeed, mutations in SLC7A9 (b^0,+^AT) lead to cystinuria (MIM 220100) [10,11], whereas mutations in SLC7A7 (y + LAT1) result in lysinuric protein intolerance (LPI) (MIM 222700) [12]. In addition, mutations in SLC7A5 and SLC7A8 (LAT1 and LAT2, respectively) cause autism-related disorders and contribute to age-related hearing loss and cataracts, respectively [13,14,15]. SLC7A11 (xCT), which mediates cystine uptake and glutamate efflux, is essential for Kaposi’s sarcoma-associated herpesvirus infection [16], and it regulates basal levels of extra-synaptic glutamate [17]. Finally, like SLC7A11, SLC7A5 is overexpressed in many human tumors, thereby suggesting that these amino acid transporters are essential for tumor cell survival and progression [18,19].

APC transporters, which comprise 12 transmembrane domains (TMs), present the APC superfamily fold (also named LeuT-fold or 5 + 5 inverted repeat fold). This fold is characterized by two structurally similar repeats, each containing a core of five consecutive TMs (TMs 1 to 5 and TMs 6 to 10), related by a pseudo-twofold symmetry axis located in the plane of the membrane [20]. As a hallmark of this fold, the first TM in each of the two inverted repeats (TM1 and 6) is discontinuous and consists of two short alpha-helices connected by a highly conserved unwound segment [20]. This structural feature is important for the translocation mechanism of APC fold transporters, as it is part of the substrate co-ordination site in solved amino acid transporters with this fold [20,21,22,23,24,25,26,27]. In addition, atomic structures of various conformational states of the same transporter, as well as those of several transporters in the APC superfamily, led to the proposal that structural changes are associated with substrate binding and translocation [21,28].

To gain insight into the molecular mechanisms associated with transporters’ substrate selectivity and translocation and to identify the molecular bases of pathogenic mutations, it is essential to have structural information about the target protein at atomic resolution. Until 2018, only prokaryotic amino acid transporters ApcT, AdiC, and GadC within the APC family had been solved [23,29,30,31,32,33]. These transporters were the closest mammalian SLC7 homologues (amino acid sequence identity (AAI) ranging from 14% to 20%) whose atomic structure had been solved, and they were regularly used as a template for the generation of atomic models of SLC7 transporters [34,35,36,37,38]. However, in the last three years, several atomic structures of bacterial and human LATs and CATs have been solved, thereby paving the way for the dissection of their molecular transport mechanisms [24,25,26,27,39,40,41,42].

## 2. Structural Information about the APC Family

The homodimeric arginine:agmatine antiporter AdiC (TC# 2.A.3.2.5) was the first structure resolved for a transporter in the APC family. All the AdiC structures reported are in an outward-facing conformation, either with or without substrate, and also substrate-bound occluded [23,30,31,32,43] (Table 1). Thus, despite its low sequence identity (around 18%), AdiC has long been used as a template for the generation of structural models of human LATs [34,35,36,37]. Indeed, these models of human transporters based on AdiC have allowed the identification of various relevant features associated with their substrate recognition and translocation mechanisms [34,35,44].

In addition to AdiC, other APC family members have been reported in different conformations. In this regard, Shaffer et al. solved the structure of the proton-coupled amino acid transporter ApcT (TC# 2.A.3.6.3) [29] in a fully occluded apo conformation. In addition, Ma et al. in 2012, published the structure of the glutamate:GABA antiporter GadC (TC# 2.A.3.7.3) [33], adopting the inward-facing apo open state—the first structure in the APC family of transporters in an inward conformation. However, in contrast to other recently resolved members of this family, the GadC structure showed a pH-regulated C-plug domain occupying most of the intracellular vestibule, thus hindering the identification of potentially relevant gating and substrate-interacting residues.

It was not until 2018 that the first SLC7 transporter was obtained. CAT subfamily member GkApcT was the first structure to be solved in an inward-facing occluded substrate-bound state [25]. The resolution of this structure, together with those mentioned above, sheds light on the translocation mechanisms of the APC family members, despite the distinctive features underlying LAT mechanisms remaining unknown. However, since 2019, several LAT structures have been solved in different conformations, thereby improving our understanding of their transport mechanisms and bringing about an updated scenario for this subfamily. Large amino acid transporter 1, LAT1 (TC# 2.A.3.8.25, SLC7A5), was solved in an inward-facing open conformation apo, and with an inhibitor (2-amino-2-norbornanecarboxylic acid; BCH) [26,27]. BCH was located in the putative substrate-binding site in a similar position as that previously described for substrate-bound structures GkApcT and AdiC, interacting simultaneously with TMs 1 and 6 [23,25,43]. Similarly, the bacterial alanine–serine–cysteine exchanger BasC (a member of the LAT subfamily) was also resolved in an inward-facing open conformation apo, and in the presence of substrate [24] co-crystallized with a nanobody. Comparison of substrate-free and substrate-bound structures, together with structural superposition with GkApcT and molecular dynamics analysis, has contributed to our understanding of the molecular bases underlying transport mechanisms in LATs [24].

Recently, inward-facing apo and substrate-bound states of human transporters LAT2 (TC# 2.A.3.8.20, SLC7A8) [42], b^0,+^AT (TC# 2.A.3.8.19, SLC7A9) [40,41], and xCT (TC# 2.A.3.8.18. SLC7A11) [46] have been solved, thus furthering our understanding of the molecular bases governing substrate recognition. All these solved structures highlight the preference of LATs to expose the substrate cavity to the cytosol rather than to the extracellular space. However, recently, human LAT1 (hLAT1) has also been solved in an outward-occluded inhibitor-bound state [39], thus providing, for the first time within the APC family, the opportunity to analyze the structural rearrangements associated with the inward to outward transition in a particular transporter protein. Finally, the most recent eukaryotic b^0,+^AT structures were successfully solved in reconstituted lipid nanodiscs [45]. This achievement marked the first APC family member to be solved in a lipidic environment, and it will contribute to the identification of particular structural adaptations associated with the biophysical features of the lipid environment.

The APC superfamily transport cycle was proposed in 2013, based on a wide variety of structurally solved APC-fold transporters [7]. Nevertheless, this cycle included transporters with low sequence identity and diverse translocation mechanisms (sodium- or proton-dependent and -independent transporters, obligatory and nonobligatory exchangers), thereby raising doubts as to whether the different transporters of the APC superfamily would necessarily adopt the distinct conformations that appeared in the aforementioned cycle. Recent advances in the structural biology of LATs are helping to decipher the structural and molecular details underlying the substrate recognition and translocation mechanisms of these molecules. The current availability of transporter structures in different conformations within the same subfamily (LAT) is contributing to the elucidation of the particular structural rearrangements associated with the outward to inward transition in LATs and the significant differences among the other members of the APC family (Figure 1).

## 3. Conformation States in the Substrate Translocation Cycle of APC Transporters

In the widely accepted alternate access model [47], secondary transporters (such as those in the APC family) undergo several conformational states to translocate the substrate across the membrane. During this transition, the transporter keeps the substrate accessible to only one side of the membrane at a given time by opening and closing different gates (Figure 1). The energetics of coupling between substrate binding and the conformational changes of the transporter are best explained by the “induced transition fit” mechanism [48]. Accordingly, initial recognition between the substrate and a nonoptimal binding site of the transporter in the ground state (open-to-in or open-to-out apo conformations; Figure 1) is required to trigger the transition to a transient state (occluded; Figure 1), where the binding site is reorganized to attain an optimum fit with the substrate. To compensate for the energy required reaching this transition state—which is necessary to continue the transport “catalysis”—the transporter will use the energy released from this binding in the transition state.

Regarding the APC fold, both TMs 1 and 6 are unwound in the center, forming two discontinuous helices named TM1a and TM1b, and TM6a and TM6b, which host the substrate-binding site [20,21]. Given the inverted 5 + 5 sequence repeats, a rocking-bundle-based substrate translocation mechanism is the accepted model for APC-fold transporters [21,49]. The mechanism describes the tilting of the bundle domain (formed by TMs 1, 2, 6. and 7) over the hash domain (TMs 3, 4, 8, and 9), which explains the major conformational changes during the inward- to outward-facing transition, using the symmetry in the inverted repeat. On the other hand, TMs 5 and 10 rearrangements have been suggested to also play a role in the substrate translocation mechanism [21,50]. Finally, TMs 11 and 12, present in LATs and CATs, are out of the inverted repeated sequence and form a V-shape at the external side of TM10 [21,49]. In theory, the same general mechanism would be expected for all members of the APC family, although the lack of a particular transporter protein in both inward- and outward-facing conformational states has been an important setback.

Taking an APC-fold transporter in inward-facing conformation as a starting point (Figure 1), the tilting of TM1a and TM6b, and the concomitant co-ordination of TMs 1, 5, and 8 are responsible for the occlusion of the substrate within the binding site from the intracellular cavity [24,25]. This conformational state has been referred to as occluded conformation in Figure 1. In obligatory exchangers, structural rearrangements associated with the transition between outward- and inward-facing states (and vice versa) require substrate binding [24]. This is not the case for ApcT (Figure 1), a nonobligatory amino acid exchanger which was solved fully occluded in the apo state [29]. The proton-coupling characteristic of ApcT can trigger the occlusion of the transporter in the absence of a substrate, in a similar way to what has been proposed for sodium-dependent APC-fold transporters, such as LeuT [20]. Nevertheless, although ApcT mediates the uptake of substrates in empty proteoliposomes, the presence of amino acid inside the proteoliposome enhances its transport activity. This observation suggests that, although ApcT can act as an amino acid exchanger, this is not its only function, in contrast to AdiC or most LATs [7,24,29,51], where the transit between the outward and inward conformations occurs only through the substrate-bound and occluded-substrate-bound states [24]. Thus, a fully occluded apo conformation would not be expected in APC transporters that act as obligatory exchangers.

Co-ordination between TMs 1, 5, and 8 is key for function in the APC superfamily of transporters. Indeed, in sodium-dependent members, one Na+ ion (named Na2) binds the unwound region of TM1 in bundle domains and TM8 in hash domains, resulting in increased substrate binding affinity and conformational transitions [22,52,53]. Similarly, in the sodium-independent LAT BasC, the Lys 154 side chain in TM5 substitutes the function of Na2 by mediating TM1–TM8 co-ordination during the translocation cycle [24]. Such mediation can also be seen in the occluded structures of GkApcT and LAT1 with a conserved Lys in this position [25,39]. In contrast, in the sodium-independent arginine/agmatine exchanger AdiC, which lacks a positively charged residue in the putative Na2 site, the substrate guanidinium group interacts with an aromatic side chain (Trp 293) in TM8 [30,43], connecting TM1 to TM8. Overall, these observations strongly suggest that the TM1–TM5 interaction is important for the stabilization of the inward-open conformation, while the inward-open to inward-occluded transition results in TM1–TM5–TM8 co-ordination. Once the substrate is occluded, TM1b and TM6a, the extracellular part of these TMs, tilt to open the binding site to the extracellular side, while a conserved bulky residue in TM6a (the thin gate) rotates to release the substrate to the extracellular medium. This conformational state has been referred to as outward-facing in Figure 1.

This basic inward to outward translocation mechanism was proposed based on the different conformational states of solved prokaryotic members of the APC family during the transport cycle [7]. Structural models of human LATs based on solved AdiC and ApcT structures allowed the identification of some key residues involved in substrate recognition and were applied for drug discovery technologies [34,35,37,38]. For instance, structural models of outward-facing human LAT1 identified more than 100 inhibitors of this transporter [34,35,38], most of which are amino acid derivatives. However, only a few of them are potent and selective, which, in part, may be due to the biased model of LAT1 based on the distant homologues AdiC and ApcT. In addition, these structural models of human LATs were also used as a template for molecular dynamics assays, thus proposing key structural rearrangements involved in the inward to outward transition [44]. Nevertheless, when the first structures for bacterial and human LATs were solved [24,26,27], significant differences were revealed when compared to the rest of the APC family members solved until then. Thus, upon substrate binding to the outward-facing apo state, the substrate-bound state evolves to an occluded state, with considerable tilt of TM6a, among other structural rearrangements [23,30,43]. In AdiC, TM6a occludes the substrate by tilting 40° [23,30] (Figure 2A), while, in LATs, this movement is less pronounced (Figure 2B), as suggested by the recently solved human LAT1 occluded outward-facing states [39]. This significant difference in the substrate occlusion in the outward-facing conformation when comparing AdiC and human LAT1 points to particular differences among the members of the APC family with respect to the structural rearrangements that occur during substrate translocation. Finally, the position of TM6a in the human LAT1 occluded outward-facing conformation (PDB ID: 7DSL) coincides with that of TM6a in the inward-facing open structures of both prokaryotic and human LAT subfamily members (Figure 2C) (Table 1). In contrast, TM6b is clearly different, reinforcing the role of TM6 in substrate occlusion, both in inward- and outward-facing conformations (Figure 2C).

Interestingly, structural superimposition of the hLAT1 inhibitor-bound outward-facing occluded state [39] and bacterial SLC7 transporter GkApcT in substrate-bound inward-facing occluded conformation [25] shows no significant differences in the position of TM1b and TM6a, thus suggesting a similar conformational state for the two solved proteins (Figure 2D,E). Given these observations, we propose that LATs only have a fully occluded conformation as an intermediate step between outward and inward open conformations. However, although this is a tempting notion, it is necessary to compare the same protein in different conformations to avoid the bias derived from the comparison of different proteins, as reported here for AdiC and hLAT1. In addition, the inhibitor-complexed structures of hLAT1, although informative, cannot represent the native substrate-bound outward-facing occluded conformation due to the particular inhibitor–transporter interactions. Thus, as is also the case for mutated versions of solved transporters, the use of inhibitor-complexed structures could lead to bias in the general rules of the translocation mechanism proposed for different transporters, especially when they are distant in evolution. Thus, solving native transporters in the presence and absence of natural substrates in various conformations will shed light on the particular structural rearrangements that govern substrate recognition and translocation in a particular group of transporter proteins.

## 4. Design of the Substrate-Binding Site

Understanding the conformational changes that occur during each transport cycle is only one of the important steps towards the elucidation of the transport mechanism. In this regard, substrate binding is another additional key aspect. In recent years, several atomic structures of human HATs [26,27,39,40,41,42,45] and bacterial LAT [24] and CAT homologues [25] have been solved in different conformations of the translocation cycle, and in the presence and absence of substrate, thus paving the way for the dissection of the molecular transport mechanisms (Table 1).

The substrate binding site of APC family members, solved with substrate-bound [23,24,25,27,30,40,42,43], and both in inward- and outward-facing conformations, shows a basic conserved design: the α-amino and carboxyl moiety of the substrate bind to unwound segments of TM1 and TM6 to interact mainly with atoms of the protein backbone. Structural comparison of solved members of the APC family reveals that the TM1a–TM1b loop is almost identical, with the conserved G (S/T/A/V)G motif in the middle, creating a loop structure [24] (Figure 3A). This configuration exposes the backbone amide of the second S/T/A/V residue to the solvent, providing a hydrogen-bond donor to one of the substrate carboxyl oxygen atoms [23,24,25,27,30,40,42,43] (Figure 3A). In addition, the electropositive potential of the helix dipole of TM1b would provide a favorable interaction with the substrate carboxyl group. In contrast, the TM6a-TM6b loop is less conserved [24]. Of note, a semi-conserved bulky residue is present in TM6 at the beginning of the unwound region and acts as an external thin gate, providing the main steric occlusion that seals the binding site from the extracellular side during transport (Figure 3A,B). Thus, the most noticeable conformational change between the outward-occluded and outward-open state occurs in TM6, which tilts away from the binding site in the outward-open conformation (Figure 2A), facing the thin external gate alternatively inward or outward, closing or opening the binding site (Figure 3B). In addition, the main chain carbonyl of the thin gate residue provides a hydrogen-bond acceptor for the substrate α-amino nitrogen (Figure 3A,B), and its hydrophobic side chain forms a hydrophobic interaction with the substrate Cβ atom, thus stabilizing the substrate-bound conformation [23,24,25,27,30,40,42,43].

Functional studies of the bacterial LAT BasC with amines or alkyl derivatives of the amino acid substrates indicate that the α-amino and carboxyl groups are necessary for proper binding and/or to trigger the transport cycle [24]. This feature parallels transport requirements in mammalian LAT1 and LAT2, where the α-amino and the α-carboxyl (or a modified carboxyl) groups are required for transport [54,55,56]. Nevertheless, small differences can be found when comparing solved APC-fold transporters. Indeed, in the CAT homologue GkApcT, a water molecule connects the substrate to backbone atoms of residues in the unwound segment of TM1 [25]. This observation suggests a slightly different substrate binding recognition between CAT and LAT subfamilies or a modification of the binding site upon occlusion [24,25]. On the other hand, the more remote APC transporter AdiC mediates the exchange of arginine and its decarboxylated form, agmatine. The particular spatial arrangement of agmatine in the substrate-binding site of AdiC allows the amino group of the substrate to interact simultaneously with the unwound regions of TM1 and TM6, thus compensating for the lack of the alpha-carboxyl group in its interaction with the unwound region of TM1 [43].

Finally, in all the resolved members of the APC family, except GadC, there is a conserved tyrosine residue in TM7 that directly interacts with the alpha-carboxyl group of the substrate [24,25,27,40,42]. This direct interaction would take place only in the inward-facing conformation since, in the case of AdiC and human LAT1, this interaction is not present in the outward-facing state (Figure 3A,B) [23,30,39,43]. The role of this connection in the inward-facing conformation has been linked to the asymmetry in apparent substrate affinity between the intracellular and extracellular binding sites present in several members of the APC family [24,51,57,58]. Indeed, the Y236F mutation in the bacterial LAT BasC resulted in a 4-fold increase in the apparent affinity of the substrate on the cytoplasmic side of the transporter, without affecting that of the extracellular side. Interestingly, Tyr 236, which is fully conserved among human LATs [24], sits in the same position as the sodium ion in the sodium-one (Na1) site in sodium-dependent APC superfamily transporters, where the cation participates in substrate binding [59,60,61].

Nevertheless, although the location of this binding site and that of the key residues involved in both substrate recognition and translocation appear to be conserved, the composition of the residues coordinating the substrates varies significantly [23,24,25,27,30,40,42,43]. In support of this notion, APC family members range from transporters with very broad substrate specificity to highly specific transporters (e.g., LAT2 compared with AdiC) [7,25,29,33,51,57]. This observation suggests that, although transporters are similar in design, substrate recognition relies on particular residues strategically located in the binding site. In this regard, functional studies with hLAT1 suggest that the lateral chain of residues in the substrate cavity determines the size of the substrates [26]. The design of the substrate cavity positions the side chain of the amino acid substrates mainly towards TM3, TM6, TM8, and TM10, where specific interactions between particular residues and the substrate side chain determine the substrate selectivity of the transporter. In this regard, in the bacterial arginine/agmatine exchanger AdiC, the substrate guanidinium group interacts through a π-cation interaction with the side chain of Trp 293 (TM8) and with the backbone of Ala 96 and side chains of Cys 97 and Asn 101 (TM3), and Ser 357 (TM10) via hydrogen bonds (Figure 3B) [23,30,43]. These interactions allow the substrate to selectively bind to the open-to-out state of the transporter, and then to stabilize the occluded conformation. Similarly, structural modeling of hLAT1 based on AdiC identified Cys 335 and Ser 342 (equivalent to Trp 293 in TM8), and Cys 407 (equivalent to Ser 357) as substrate interactors, thus suggesting a similar binding site design [34,38]. In addition, mutation K295C in the bacterial Ser/Thr exchanger SteT and M321S in the bacterial CAT GkApcT, both residues homologous with the main substrate attractor residue Trp 293 in AdiC, markedly broaden the amino acid substrate selectivity profile [25,62].

Recent structures of human LAT1, LAT2, and b^0,+^AT have shed light on the molecular determinants of substrate selectivity in LATs. Comparison of the binding sites of these transporters and the AdiC structures revealed significant differences. Indeed, residues enclosing the AdiC binding site are replaced by smaller residues in LAT1, creating a larger volume for substrate binding, e.g., M104, I205, and W293 in AdiC correspond to the smaller V148, G255, and S342 in LAT1 [26,27,39]. In addition, a second cavity close to the substrate-binding site and defined mainly by residues from TMs 6 and 10, which could be relevant in the recognition of the substrate side chain, has also been defined for the human counterparts [26,27,40,41,42]. However, none of the inward-facing substrate-bound solved structures of these proteins show substrate molecules in this cavity, thereby suggesting that it could be occupied in another conformation during the transport cycle. In fact, mutations of the residues that shape this cavity result in a reduction in transport activity, thus corroborating their relevance for the correct functioning of the transporter [27,40,42]. This second cavity has not been observed in any of the prokaryotic transporters of the APC family resolved to date [23,24,25,29,30,32,33,43], thereby pointing to significant differences in the conformation of the substrate-binding site during the translocation cycle between the various members of this family.

The functional and structural characterization of pathogenic mutants responsible for human aminoacidurias has revealed specific residues involved in transport activity in humans [63,64,65]. In this regard, when mutated to methionine (T123M), residue T123, located in TM3 of b^0,+^AT within the putative substrate-binding pocket, causes isolated cystinuria without urine hyperexcretion of dibasic amino acids [40,66]. This observation would suggest that this residue is involved in the specific recognition of cystine but not arginine, lysine and ornithine, and that its mutation to methionine differentially affects the recognition of cystine and dibasic amino acids in b^0,+^AT. In this regard, recent resolution of the human b^0,+^AT transporter shows that this residue does not interact directly with arginine in inward-facing conformation [40]. However, the absence of the transporter structure in the presence of cystine precludes the identification of the molecular basis of Thr 123 in cystine selectivity and its role in cystinuria. Similarly, the lysine to glutamate change, a mutation that, in human y+LAT1 (K191E), causes lysinuric protein intolerance [65], also dramatically reduces transporter activity [24,65]. This lysine residue, which is conserved in all members of the LAT and CAT subfamilies [7,24,67], has been reported to play a key role in the release of the substrate to the cytosol and in facilitating the outward-to-inward transition [24], thereby increasing the apparent affinity for substrates at the extracellular face.

A physiologically relevant characteristic of LATs is asymmetry in the apparent substrate affinity at both the intracellular and extracellular sides of the transporter. This asymmetry allows LATs to regulate intracellular amino acid pools (mM concentrations) by exchange with external amino acids (µM concentration range) [58]. Interestingly, most of the atomic structures of LAT subfamily members were solved in very similar inward-facing conformations (overall RMSD < 1.5 Å) (Table 1). This high conformational pre-eminence would represent the most “stable” state, which may also be related to the physiological asymmetry of the apparent substrate affinity at the two sides of the membrane. The recent resolution of the inward- and outward-facing structures of hLAT1 will allow identification of the extracellular determinants of substrate selectivity and, thus, the establishment of whether, as with apparent affinity, there is also asymmetry in the substrate selectivity profile.

## 5. Conclusions

The wealth of structure/functional knowledge acquired with transporters with APC fold can now start to be translated to the members of the LAT subfamily. Present structures cover inward-facing apo and substrate-bound, as well as variations of the occluded substrate-bound states. Outward-facing conformations of LATs are needed to close the structural biology of LATs. Still, residues involved in substrate binding to the inward-facing conformation and determinants of the asymmetric interaction with substrates at both sides of the plasma membrane can start to be identified. Substrate recognition in the outward-facing and occluded states is the next objective to close the transport cycle of LATs.

## Figures and Tables

**Figure 1 membranes-11-00602-f001:**
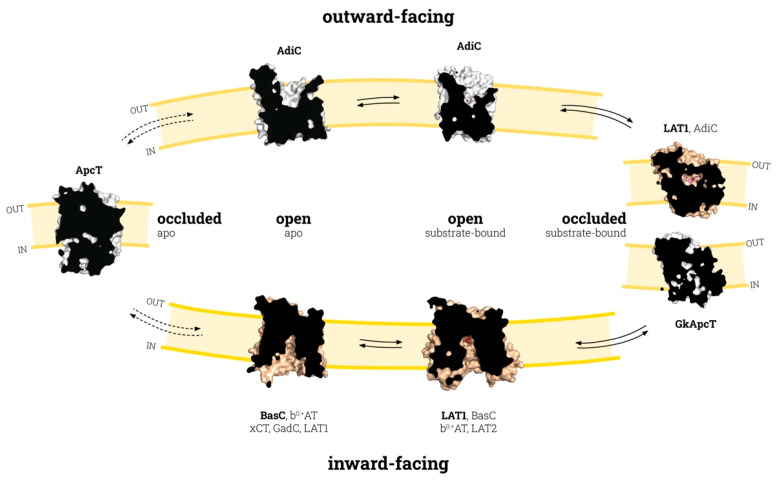
Conformational states of the APC family transport cycle. The transition between inward- and outward-facing states involves the occluded substrate-bound state (bold arrows), except for the transporters with a nonobligatory exchange activity, which could also explore the occluded apo state (dashed arrows). Each conformational state is shown by a sliced-surface representation of the transporter indicated in bold, colored gold for LATs and white for other APC transporters. All transporters with known structures in each state are indicated. PDB codes for the structures depicted here are the following: ApcT occluded apo (3GIA), AdiC outward-facing open apo (3NCY), AdiC outward-facing open substrate-bound (5J4N), human LAT1 and GkApcT outward-facing substrate-bound occluded (7DSK and 5OQT, respectively), human LAT1 inward-facing open substrate-bound (6IRT), and BasC inward-facing open apo (6F2G). The two occluded substrate-bound structures shown (hLAT1 and GkApcT) might represent small variations within the occluded substrate-bound state of LATs/CATs. Protein representations were created with PyMOL software (The PyMOL Molecular Graphics System, Version 2.3, Schrodinger, LLC. 2010).

**Figure 2 membranes-11-00602-f002:**
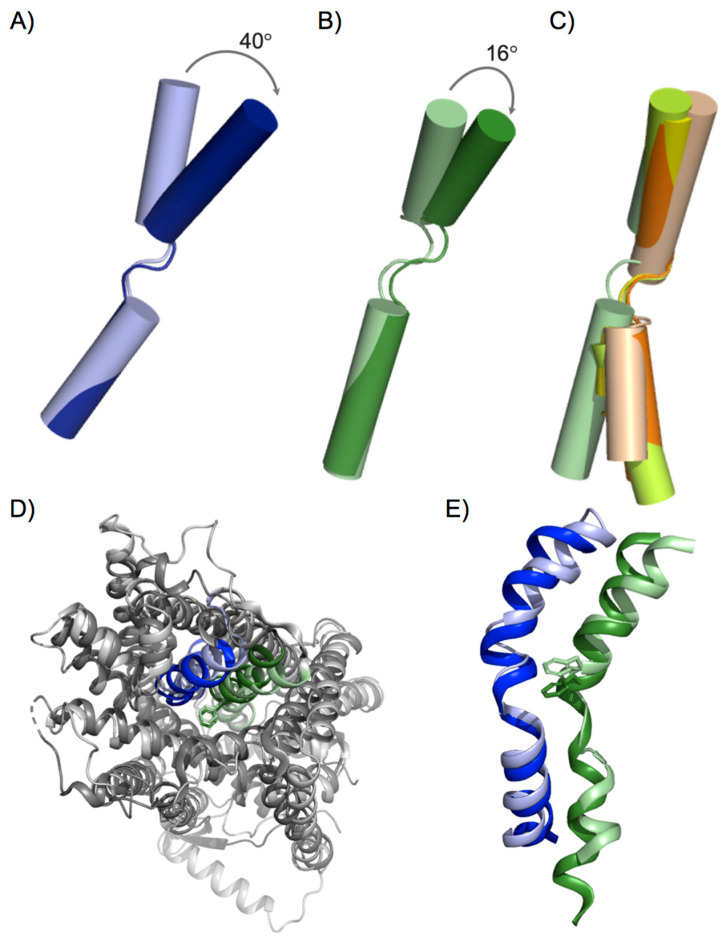
TM1 and TM6 structural rearrangements upon substrate occlusion. Structural superimposition of transmembrane domain 6 of (**A**) AdiC substrate-bound outward occluded (dark blue, PDB ID: 3L1L) and outward open (light blue, PDB ID: 5J4N) structures; (**B**) human LAT1 outward occluded, complexed with either JX-078 (pale green, PDB ID: 7DSL) and/or 3,5-diiodo-L-tyrosine (dark green PDB ID: 7DSQ); and (**C**) human LAT1 outward occluded with JX-078 (pale green), human LAT1 inward-open (lemon green, PDB ID: 6IRT), human b^0,+^AT (orange, PDB ID: 6LI9), and BasC (salmon pink, PDB ID 6F2G). (**D**) Upper extracellular view of the structural superimposition of GkApcT substrate-bound inward-facing occluded conformation (solid colors, PDB ID: 5OQT) and human LAT1 outward occluded, complexed with JX-078 (light colors, PDB ID: 7DSL). (**E**) Detailed structural superimposition of transmembrane domains 1 (blue) and 6 (green) from (**D**). Protein representations were created with PyMOL software (The PyMOL Molecular Graphics System, Version 2.3, Schrodinger, LLC. 2010).

**Figure 3 membranes-11-00602-f003:**
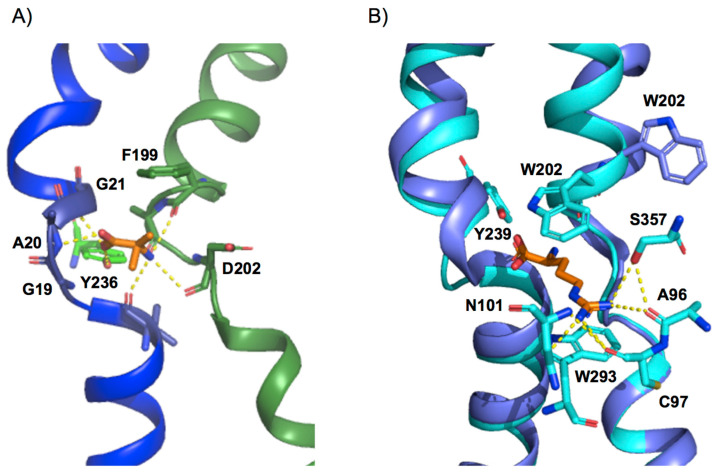
Substrate-binding site design in APC transporters. (**A**) View of the bacterial alanine–serine–cysteine exchanger substrate-binding site (PDB ID 6F2W). Distances between atoms of the substrate (AIB, orange) and BasC residues compatible with H bonds are indicated (dashed lines). F199 acts as the external thin gate, sealing the binding site from the extracellular side. (**B**) Superimposition of AdiC outward-facing open apo (PDB ID 3J4I) and outward-facing arginine-bound and occluded (PDB ID 3L1L). Distances between atoms of the substrate side chain (L-arginine, orange) and AdiC residues compatible with H bonds are indicated (dashed lines). Tyr 239 (equivalent to Tyr 236 in BasC) is also depicted. Trp 202, which acts as the external thin gate, tilts away from the binding site in the outward-facing open conformation, allowing substrate release. Protein representations were created with PyMOL software (The PyMOL Molecular Graphics System, Version 2.3, Schrodinger, LLC. 2010).

**Table 1 membranes-11-00602-t001:** Current APC family members with structure solved. Transporters’ abbreviated and complete names, TCDB, and/or SLC classification and corresponding species are indicated. Structures are grouped by conformational states indicating the substrate or inhibitor bound. The amphipathic compound used (Amph.) in the last step of the purification process, the overall resolution (Å), Protein Data Bank (PDB) entry codes, and notes (mutations and chaperones used) are also indicated in the structural information columns.

Transporter		Conformational State			Structural Information				Ref.
Abbr. & Class	Name and Spp.	Facing	Opening	Substrate	Amph.	Å	PDB	Notes	
AdiC TC# 2.A.3.2.5	Arginine:agmatine antiporter (*E. coli*)	Outward	Open	-	NG	3.61	3LRB		[32]
				-	NG	4.00	3LRC		[32]
				-	DM	3.20	3NCY	(a)	[31]
				-	NG	2.21	5J4I	N101A	[43]
				L-Arg	Cymal-6	3.00	3OB6		[23]
				Agmatine	NG	2.59	5J4N		[43]
			Occluded	L-Arg	NG	3.00	3L1L	N22A	[30]
ApcT TC# 2.A.3.6.3	Proton coupled amino acid transporter (*M. jannaschii*)	Inward	Occluded		OG	2.32	3GIA		[29]
					OG	2.59	3GI8	K158A (a)	[29]
					OG	2.48	3GI9	(a)	[29]
GadC TC# 2.A.3.7.3	Glutamate:GABA antiporter (*E. coli*)	Inward	Open		NG + LDAO	3.10	4DJK		[33]
					NG + LDAO	3.19	4DJI		[33]
GkApcT TC# 2.A.3.3.n	Proton-coupled amino acid transporter (*G. kaustophilus*)	Inward	Occluded	L-Arg	DDM	3.13	6F34	(b)	[25]
				L-Ala	DDM	2.86	5OQT	(b)	[25]
BasC TC# 2.A.3.8.n	Ala-Ser-Cys antiporter (*Carnobacterium* sp. *AT7*)	Inward	Open		DM	2.92	6F2G	(c)	[24]
				2-AIB *	DM	3.40	6F2W	(c)	[24]
LAT1 SLC7A5 TC# 2.A.3.8.25	L-type amino acid transporter 1 (*H. sapiens*)	Inward	Open	BCH *	Digitonin	3.50	6IRT	A36E	[27]
					GDN	3.30	6IRS	A36E	[27]
					Digitonin	3.31	6JMQ	(a)	[26]
		Outward	Occluded	JX-075 *	GDN	2.90	7DSK		[39]
				JX-078 *	GDN	2.90	7DSL		[39]
				JX-119 *	GDN	3.10	7DSN		[39]
				Diiodo-Tyr *	GDN	3.40	7DSQ		[39]
b^0,+^AT SLC7A9 TC# 2.A.3.8.19	b^0,+^-type amino acid transporter 1 (*H. sapiens* and *Ovis* sp.)	Inward	Open	L-Arg	GDN	2.30	6LI9		[40]
					GDN	2.70	6LID		[40]
					Digitonin	2.90	6YUP		[41]
					Digitonin	3.40	6YV1		[41]
					Nanodisc	unreleased			[45]
					Nanodisc	unreleased			[45]
LAT2 SLC7A8 TC# 2.A.3.8.20	L-type amino acid transporter 2 (*H. sapiens*)	Inward	Open	L-Trp	GDN	2.90	7CMH		[42]
				L-Leu	GDN	3.40	7CMI		[42]
xCT SLC7A9 TC# 2.A.3.8.19	Cystine:glutamate antiporter (*H. sapiens*)	Inward	Open		Digitonin	6.20	7CCS		[46]

* Inhibitor/(a) Fragment antigen-binding/(b) MgtS transmembrane interaction/(c) Nanobody.

## Data Availability

Not applicable.

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
