# Peer review of "Rush Hour of LATs towards Their Transport Cycle"

_membranes, 2021, doi:10.3390/membranes11080602_

Round 1

Reviewer 1 Report

In this manuscript, Nicolàs-Aragó and co-authors introduce a review on the recent developments regarding the structural characterization of mammalian amino acid transporters which belong to SLC7 family, with a special focus on L-amino acid transporters (LATs) and the cationic amino acid transporters (CATs), all members of the amino acid, polyamine and organocation (APC) family of transporters. The topic of the review is of high relevance, as until 2018 only prokaryotic amino acid transporters within the APC family had been resolved structurally; the review covers the progress made in the last four years, highlighting the efforts made to solve the structures of human LATs and CATs.

The review is clear and very well written, integrating new and important data. There are some issues that the authors need to address before the manuscript can be accepted for publication.

  • A more detailed explanation regarding the delimitation between LATs and CATs in terms of substrate selectivity, inhibitors, etc., would be a plus.
  • Table 1: please add a column indicating the species corresponding to the transporter solved.
  • Figures 1-3: please indicate the software used to depict the 3D representations of the membrane proteins/fragments shown (or else, indicate the source).
  • Line 31 (minor): replace „know” with „known”.

Author Response

Comments and Suggestions for Authors. Reviewer 1.

In this manuscript, Nicolàs-Aragó and co-authors introduce a review on the recent developments regarding the structural characterization of mammalian amino acid transporters which belong to SLC7 family, with a special focus on L-amino acid transporters (LATs) and the cationic amino acid transporters (CATs), all members of the amino acid, polyamine and organocation (APC) family of transporters. The topic of the review is of high relevance, as until 2018 only prokaryotic amino acid transporters within the APC family had been resolved structurally; the review covers the progress made in the last four years, highlighting the efforts made to solve the structures of human LATs and CATs.

The review is clear and very well written, integrating new and important data. There are some issues that the authors need to address before the manuscript can be accepted for publication.

  • A more detailed explanation regarding the delimitation between LATs and CATs in terms of substrate selectivity, inhibitors, etc., would be a plus.

Reviewer 1 is right and, as suggested, more detailed information regarding differences between LATs and CATs has been added (pages 1 and 2)

  • Table 1: please add a column indicating the species corresponding to the transporter solved.

Information has been added as suggested by Reviewer 1.

  • Figures 1-3: please indicate the software used to depict the 3D representations of the membrane proteins/fragments shown (or else, indicate the source).

Information requested by Reviewer 1 has been added to the Figure Legends

  • Line 31 (minor): replace „know” with „known”.

Changed as requested by Reviewer 1

Reviewer 2 Report

In this paper Nicolàs-Aragób and colleagues reviewed the wealth of knowledge regarding the structure, function and transport cycle of LATs. Overall, the paper is nicely written, informative, and paves the next steps to resolve the structural biology of LATs

The manuscript is well organized and the cartoons are very useful to render it clearer to the reader.

It would be interesting to know if the Authors have considered to extend the discussion regarding the relevance of these transporters in various disease, including cancer, as well as their emerging role as potential drug delivery systems and/or targets of anti-cancer therapies.

and anti-tumor target therapies.

A minor issue is to use maintain the same expression when referring to LATs, CATs, APC through the manuscript. See for instance: LATs, at line 19 (abstract) and line 38 (introduction); CATs, at line 11 (abstract)and line 38 (introduction); APC, at line 15 (abstract) and line 40 (introduction).

There are a few places with grammatical errors/spelling errors.

In my opinion, this paper is interesting and nicely written and may be worthy of publication in Membranes.

Author Response

Comments and Suggestions for Authors. Reviewer 2.

In this paper Nicolàs-Aragó and colleagues reviewed the wealth of knowledge regarding the structure, function and transport cycle of LATs. Overall, the paper is nicely written, informative, and paves the next steps to resolve the structural biology of LATs

The manuscript is well organized and the cartoons are very useful to render it clearer to the reader.

It would be interesting to know if the Authors have considered to extend the discussion regarding the relevance of these transporters in various disease, including cancer, as well as their emerging role as potential drug delivery systems and/or targets of anti-cancer therapies and anti-tumor target therapies.

Good point raised by Reviewer 2. Although is true that the recent advances in the pathophysiology of LAT transporters have identified additional roles for this subfamily of transporters in a wide range of diseases, we wanted to focus nearly exclusively on the structural and mechanistic features recently discovered. In our opinion there are a number of recent excellent reviews on the pathophysiology of LATs, but a review integrating the increasing number of structures and the mechanistic information inferred from them was lacking. This is why we think that an extended view including transporters relevance on disease would have resulted in a loss of the main focus of this article.

A minor issue is to use maintain the same expression when referring to LATs, CATs, APC through the manuscript. See for instance: LATs, at line 19 (abstract) and line 38 (introduction); CATs, at line 11 (abstract)and line 38 (introduction); APC, at line 15 (abstract) and line 40 (introduction).

A suggested by Reviewer 2, nomenclature has been homogenized.

There are a few places with grammatical errors/spelling errors.

Spelling and grammar corrector has been applied throughout the document.

In my opinion, this paper is interesting and nicely written and may be worthy of publication in Membranes.